# Dynamics in typewriting performance reflect mental fatigue during real-life office work

**Marlon de Jong** [1¤], **Anne M. Bonvanie**[2], **Jacob Jolij** [1,3¤], **Monicque M. Lorist**[1,4¤]\*

**1** Department of Experimental Psychology, Faculty of Behavioral and Social Sciences, University of Groningen, Groningen, The Netherlands, **2** Department of Operations, Faculty of Economics and Business, University of Groningen, Groningen, The Netherlands, **3** Department of Research Support, Faculty of Behavioral and Social Sciences, University of Groningen, Groningen, The Netherlands, **4** Department of Biomedical Sciences of Cells and Systems, Cognitive Neuroscience Center, University Medical Center Groningen, University of Groningen, Groningen, The Netherlands

¤ Current address: Faculty of Behavioral and Social Sciences, University of Groningen, Groningen, The Netherlands
\* m.m.lorist@rug.nl

**Data Availability Statement:** Data cannot be shared publicly because participants did not give their consent for sharing the data with other parties than the researchers. Data are available from the Behavioral and Social Sciences Institutional Data

## Abstract

Mental fatigue has repeatedly been associated with decline in task performance in controlled situations, such as the lab, and in less controlled settings, such as the working environment. Given that a large number of factors can influence the course of mental fatigue, it is challenging to objectively and unobtrusively monitor mental fatigue on the work floor. We aimed to provide a proof of principle of a method to monitor mental fatigue in an uncontrolled office environment, and to study how typewriting dynamics change over different time-scales (i.e., time-on-task, time-of-day, day-of-week). To investigate this, typewriting performance of university employees was recorded for 6 consecutive weeks, allowing not only to examine performance speed, but also providing a natural setting to study error correction. We show that markers derived from typewriting are susceptible to changes in behavior related to mental fatigue. In the morning, workers first maintain typing speed during prolonged task performance, which resulted in an increased number of typing errors they had to correct. During the day, they seemed to readjust this strategy, reflected in a decline in both typing speed and accuracy. Additionally, we found that on Mondays and Fridays, workers adopted a strategy that favored typing speed, while on the other days of the week typing accuracy was higher. Although workers are allowed to take breaks, mental fatigue builds up during the day. Day-to-day patterns show no increase in mental fatigue over days, indicating that office workers are able to recover from work-related demands after a working day.

## Introduction

In order to be able to interact with the dynamically changing world around us, we continuously adapt our behavior. During prolonged task performance, however, this adaptation is often insufficient to counter for the increasing demands placed on our information processing system. This typically is reflected as a decline in task performance over time: people perform more slowly, make more mistakes, and are less able to correct for these mistakes [1, 2]. This

Access for researchers who meet the criteria for access to confidential data (research-data-bss@rug.nl).

**Funding:** This study, part of SPRINT@Work, is part-financed by the European Regional Development Fund, the province and municipality of Groningen, and the province of Drenthe (Grant No: T-3036, 2013). MJ, AB and ML were financed by this funding. The funders did not play a role in the study design, data collection and analysis, decision to publish, or preparation of the manuscript. There was no additional external funding received for this study.

**Competing interests:** The authors have declared that no competing interests exist.

decline in performance, commonly known as mental fatigue, occurs in many settings, and might not only have implications for productivity, but also for the safety of employees and their environment [3, 4]. More specifically, long working hours and experiencing work-related fatigue have shown to be predictors of health complaints and absence due to sickness [5, 6]. Additionally, there is an increased risk of fatigue-related accidents when employees engage in traffic after a long day of work, thereby endangering not only themselves but also others [7]. Considering the impact of mental fatigue in the working environment, it is important to conduct research on how to detect, deal with or even prevent the effects of mental fatigue.

Ever since the beginning of the 19th century researchers have studied mental fatigue and its effects on performance in controlled experimental settings and in real-life situations, such as the workplace [8–11]. Several theories regarding the cognitive mechanisms behind its manifestation have been proposed, eventually leading to the widely accepted theory suggesting that mental fatigue develops as a result of a cost-benefit evaluation of effort [12, 13]. According to this theory, if the costs of performing a task exceed the benefits of finishing the task, people will come to experience subjective feelings of mental fatigue (e.g., aversion against task performance, low vigilance) and performance deteriorates (i.e., people become slower and less accurate).

Kreapelin was the first to attempt to quantify the course of mental fatigue during task performance. It soon became clear, however, that there was no such thing as a typical decline in performance over time. Mental fatigue and its effects on behavior depend on several personal and environmental factors. For example, people are able to overcome the effects of mental fatigue if they are sufficiently motivated when they receive a monetary reward based on their performance [14] or if they drink a cup of (caffeinated) coffee [15]. Given that it is hard to define a specific course of mental fatigue over time, and employees themselves are poor at detecting when they are not capable of performing a task at an adequate level anymore [16], it is challenging to effectively monitor and prevent mental fatigue in the working environment. In order to detect this decline in performance, it is necessary to continuously monitor behavior dynamics without interference of work.

Developments in information technology, however, have made it possible to monitor behavior in novel ways, without interfering with regular work activities. For example, Pimenta and colleagues [17] developed a method for non-invasive measurement of mental fatigue by monitoring a very common behavior for office workers: typewriting performance. They found that several markers of typing performance were susceptible to the effects of time-of-day. To validate whether changes in these markers were due to mental fatigue specifically, Jong and colleagues [18] conducted an experiment in which brain activity using electroencephalography (EEG) was recorded during a 2-hour typewriting task. They were specifically interested in the P3 brain potential of which the amplitude has been known to decrease with increasing mental fatigue [19–21]. The study showed that both typing speed, reflected in the time between two subsequent keypresses (interkey interval), and typing accuracy, reflected in overall backspace use and incorrectly typed words, declined with prolonged task performance. Moreover, these deteriorations in typewriting performance with time-on-task correlated with neural makers signaling mental fatigue, indicating that monitoring typewriting markers can provide information about the level of mental fatigue, at least in a controlled setting.

Although changes in typewriting have been found to reflect mental fatigue under these standardized conditions, there are many other variables that could influence behavior dynamics under less controlled conditions. For instance, at the workplace, where deteriorations in task performance are particularly problematic, the effects of time-of-day and day-of-week have found to influence performance, as well. While time-on-task effects have mostly been studied in experimental settings, studies concerning the effects of time-of-day and day-of-week are

generally performed in real-life settings, investigating self-reports. A study of Linder and colleagues [22], for example, showed that clinicians prescribed unnecessary antibiotics more often in the afternoon as compared to the beginning of the day. Similar effects have been found over the different days of the week, where employees have been found to feel more energized after the weekend, resulting in better reported performance at the beginning compared to the end of the week [23]. Although these effects work on different time-scales, they all resulted in changes in (self-reported) performance levels. Moreover, continuously performing a task, and engaging in work for multiple hours or days, requires rest to restore performance to its former level [24]. More specifically, time-on-task effects can be reversed by taking a short (coffee) break [25], time-of-day effects can be reversed by a nights rest [26], and day-of-week effects can be reversed by a weekend break [23]. Although there is substantial evidence that prolonged task performance, manipulated by time-on-task, time-of-day, and day-of-week, separately influence performance, interestingly, it is not yet known how these factors interact and subsequently influence behavior dynamics.

Previous experimental studies on mental fatigue mainly focused on the effects of time-on-task on behavioral performance, investigating isolated effects of prolonged task performance on specific cognitive processes (e.g., error processing [2]). In addition, studies in real-life settings focused on specific professions, especially those involving shift-work [27, 28], where the manifestation of fatigue was expected to be potent or even dangerous, given its relationship with serious accidents [29, 30]. There seems to have been done little research on the manifestation of mental fatigue during regular 9 to 5 jobs.

### Present study

In order to gain more insight in the manner in which behavioral dynamics in the workplace are influenced by time-on-task, time-of-day and day-of-week, we first focused on validating a potentially useful method to study mental fatigue on the work floor without interfering with regular working activities. To this end, markers in typewriting that were found to be sensitive to mental fatigue in a lab setting (i.e. interkey interval and backspace use) were recorded for six consecutive weeks during regular office work[16]. Second, we investigated the influence of mental fatigue on these markers at different time-scales (i.e., time-on-task, time-of-day, and day-of-week).

In line with findings in an experimental setting [18], we hypothesized that there would be a main effect of time-on-task on both the interkey interval and the percentage of backspaces, where we expected that both measures would increase with time-on-task. Secondly, we hypothesized that the magnitude of the effect of time-on-task on these performance measures would depend on time-of-day (main effect). That is, we expected a larger increase in both the interkey interval and the percentage of backspaces with time-on-task in the afternoon (interaction) than in the morning. Lastly, we hypothesized that typewriting patterns would change over the course of the week. We expected these changes to manifest in two ways. First, we hypothesized that employees became slower and less accurate over the week (main effect), and second, we expected a larger decline of performance (interkey interval and backspace use) with time-on-task over the week (interaction).

## Materials and methods

### Participants

Forty-five office workers gave their written informed consent to participate in a study that was approved by the Ethics Committee of the Faculty of Economics and Business in Groningen. This research complied with the tenets of the Declaration of Helsinki. Participants were

employees of the Faculty of Economics and Business of the University of Groningen and were recruited via the health and safety coordinator of the faculty. They were included if they worked for at least 0.8 full-time equivalent (32 h a week) and typewriting activities were part of their work. Only datasets which contained more than 30 subsets of more than 45 min of continuous typing were included in order to perform reliable statistical analyses. From now on we will refer to these subsets as tasks. As a result, data of 23 employees was excluded from the analysis, leaving data of 22 employees (12 females, *M* = 48.1 year, *SD* = 13.4). There was variation in function profile across participants that were included in the study (i.e., scientific staff, support staff). Participants that were excluded from the analyses performed working activities during the measurement period that did not include the required amount of typing activities (e.g., teaching and collecting research data), which was specifically the case for Ph.D. students and (Postdoctoral) researchers. In addition, a number of participants worked on multiple workstations during the 6 weeks of data collection, which was reflected in a limited amount of typewriting data that was recorded from these participants at the workstation on which the recording software was installed. Data of these participants were excluded from the analyses, as well.

## Apparatus and materials

The experiment was conducted in the natural working environment of the participants at the faculty of Economics and Business of the University of Groningen. The experimental setup consisted of an office chair behind an adjustable desk, a windows computer with a QWERTY keyboard, and screen support. The working environment was adjusted according to the occupational health and safety guidelines of the faculty. The percentage of backspaces and the inter-key interval was acquired using keylogging software (aXtion).

## Typing performance

Previous research of de Jong et al. [18], found backspace use and the interkey interval to be susceptible to the effects of mental fatigue in a controlled lab setting. In order to monitor these typing indices, keylogging software, installed on the workstations, registered a timestamp at the start of each keystroke. To safeguard the confidentiality of the typed text during the study, only the backspace key was given a unique marker. Each minute, the average interkey interval (the time between two subsequent keystrokes) and the percentage of backspaces of the preceding 15 min was calculated and registered for offline analysis. If the time between two subsequent keystrokes was longer than 5 s, the interkey interval was not included in the average. A series of average values was included in subsequent analysis if more than 45 successive averages were recorded. In the present study, continuous typewriting was defined as typewriting during a block of at least 45 minutes.

## Procedure

Typing performance was monitored for 6 weeks in the natural working environment of the participants. Data collection of the first cohort started on the first Monday of May and the second cohort started on the first Monday of November. A week before the start of the monitoring period, the keylogging software was installed on the computers of the participants and the office environment was confirmed to be or adjusted according to occupational health and safety guidelines of the faculty. During this week, participants also filled out a questionnaire with demographic and work-related questions (S1 Appendix). Each Monday, starting in the second week of the experiment, participants filled out a questionnaire with general questions about how they experienced the week before (S2 Appendix). Each working day, participants

received real-time feedback on their performance provided via text messages on their mobile phones and via email. An overview was provided via email at the end of the day.

## Statistical analysis

Statistical analysis was conducted in R version 3.4.4 [31, 32]. For statistical significance testing, we used a mixed-modelling approach using the lme4 package version 1.1–21 [33]. The package lmertest version 3.0–1 was used to obtain statistical significance by approximating the degrees of freedom using the Satterthwaite approximation [34]. The data provided to the models included the interkey interval and the percentage of backspaces. The models contained a varying intercept per participant. In addition, a varying slope for time-on-task and time-of-day by subject was added to the model if the fit of the model improved as indicated by the Akaike Information Criterion (AIC) [35]. The models used to statistically test the effects of time-on-task (120 min of continuous typewriting), time-of-day (morning and afternoon), and day-of-week (Monday, Tuesday, Wednesday, Thursday, and Friday) on the dependent typewriting variables (i.e., interkey interval and percentage of backspaces) are listed in Table 1.

Post-hoc tests were performed to assess the main and interaction effects, adjusting error rates according to Bonferroni. First, to estimate the difference between the effect of time-on-task between the morning and the afternoon, polynomial contrasts were compared using pairwise comparisons. Second, pairwise comparisons were administered to compare the interkey interval and backspace use on the different days of the week. Finally, polynomial contrasts were used to estimate the linear and quadratic trends with time-on-task in the morning and the afternoon, and over the different days of the week. Statistical tests were considered significant at $p < .05$.

**Speed and accuracy.** In order to investigate the relationship between typing speed (interkey interval) and typing accuracy (backspace use) during continuous typewriting, we calculated the regression coefficients that described the effect of time-on-task on the dependent variables in the morning and in the afternoon for each participant. For these personalized regression coefficients, we calculated Pearson's correlations to identify whether changes in typing speed and accuracy were related.

## Results

In order to systematically discuss the results, we first report the effects of time-on-task on the interkey interval, reflecting typing speed, and backspace use, reflecting accuracy. Thereafter,

**Table 1. Models used to statistically test the effects of time-on-task, time-of-day, and day-of week on the interkey interval and the percentage of backspace keystrokes.**

| Dependent variable | Equation |
|---|---|
| Interkey interval$_n$ | $\beta_{0,j} \quad + \beta_1\, time\ on\ task_n + \beta_2\, time\ of\ day_n$ |
| and | $+ \beta_3\, day\ of\ week_n$ |
| Backspace use$_n$ | $+ \beta_5\, time\ on\ task_n \times time\ of\ day_n$ |
| | $+ \beta_6\, time\ on\ task_n \times day\ of\ week_n$ |
| | $+ \beta_7\, time\ of\ day_n \times day\ of\ week_n$ |
| | $+ \beta_8\, time\ on\ task_n \times time\ of\ day_n$ |
| | $\times day\ of\ week_n + \epsilon_n$ |

$n$ reflects a time-block (minute) and $j$ reflects a participant. $\beta_0$ reflects the intercept of the model, $\beta_{1-8}$ reflect the regression coefficients, and $\epsilon$ reflects the error term. The notation for these models allowed for a varying intercept per participant (as indicated by $j$).

**Table 2. The main and interaction effects of time-on-task, time-of-day, and day-of week on the interkey interval.**

| Main and interaction effects | F-value | Dfs | p-value |
|---|---|---|---|
| Time-on-task$^2$ | 17.75 | 2, 94707 | < .001 |
| Time-of-day | 1.07 | 1, 94719 | .302 |
| Day-of week | 74.15 | 4, 94707 | < .001 |
| Time-on-task$^2$ × time-of-day | 6.90 | 2, 94703 | .001 |
| Time-on-task$^2$ × day-of week | 47.57 | 8, 94701 | < .001 |
| Time-of-day × day-of week | 54.69 | 4, 94705 | < .001 |
| Time-on-task$^2$ × time-of-day × day-of week | 72.08 | 8, 94701 | < .001 |

we go into the effects of time-of-day and the interaction of time-of-day with time-on-task on the same measures. Lastly, we report how these typewriting patterns change over the different days of the week. The models that were used to statistically test the effects of prolonged task performance on the different time-scales (i.e., time-of-day, time-of-day, and day-of-week) can be found in Table 1. An overview of the main and interaction effects is provided in Tables 2 and 3, respectively.

## Time-on-task

The results showed that both the interkey interval ($F(2, 94707) = 17.75$, $p < .001$) and the percentage of backspace keystrokes ($F(2, 91637) = 284.28$, $p < .001$) changed with prolonged task performance (i.e., subset of > 45 minutes of continuous typewriting). That is, in general, we observed an increase in both the interkey interval and the percentage of backspaces, reflecting a decrease in typing speed and a decline in typing accuracy with time-on-task. However, as expected, these effects were modulated by time-of-day and day-of-week. These modulations will be discussed below.

## Time-of-day

Although mean interkey interval (main effect **time-of-day:** $F(1, 94719) = 1.07$, *n.s.*) did not differ between the morning and the afternoon, the effect of time-on-task on the interkey interval was modulated by time-of-day (interaction effect **time-on-task × time-of-day**: $F(2, 94703) = 6.90$, $p = .001$; **afternoon** *minus* **morning**$_{linear}$: $z = 3.29$, $p = .006$; **afternoon** *minus* **morning**$_{quadratic}$: $z = -3.51$, p = .002). That is, post-hoc tests revealed that the interkey interval remained stable during continuous typewriting in the morning, but in general it increased with 11.6 ms during two hours of continuous task performance in the afternoon (see Table 4 and Fig 1A).

Backspace use increased from the morning to the afternoon (main effect **time-of-day:** $F(1, 91656) = 36.55$, $p < .001$). Additionally, the effect of time-on-task on backspace use differed

**Table 3. The effects of time-on-task, time-of-day and day-of week and their interaction on the percentage of backspaces described by F-test.**

| main and interaction effects | F-value | Dfs | p-value |
|---|---|---|---|
| Time-on-task$^2$ | 284.28 | 2, 91637 | < .001 |
| Time-of-day | 36.55 | 1, 91656 | < .001 |
| Day-of week | 53.37 | 4, 91637 | < .001 |
| Time-on-task$^2$ × time-of-day | 12.93 | 2, 91634 | < .001 |
| Time-on-task$^2$ × day-of week | 21.85 | 8, 91633 | < .001 |
| Time-of-day × day-of week | 36.93 | 4, 91635 | < .001 |
| Time-on-task$^2$ × time-of-day × day-of week | 9.54 | 8, 91633 | < .001 |

**Table 4. The effect of time-on-task on the interkey interval and backspace use in the morning and the afternoon.**

| Dependent variable | Time-of-day | Polynomial | z-value | $M_{change}$ (SE) |
|---|---|---|---|---|
| **Interkey interval (ms)** | Morning | Linear | -1.92 | 2.94 (1.53) |
| | | Quadratic | 1.73 | |
| | Afternoon | Linear | **5.42**\*\*\* | 11.51 (2.12) |
| | | Quadratic | **-3.07**\* | |
| **Backspace use (% of backspace keystrokes)** | Morning | Linear | **19.74**\*\*\* | 1.57 (0.08) |
| | | Quadratic | **-11.11**\*\*\* | |
| | Afternoon | Linear | **14.99**\*\*\* | 1.63 (0.11) |
| | | Quadratic | **-9.84**\*\*\* | |

$M_{change}$ reflects the average change in the dependent variable from the 1st to the 120th minute of continuous typewriting.

Bolded values are significant

* $p < .05$;

** $p < .01$;

*** $p < .001$

between the morning and the afternoon (interaction effect **time-on-task × time-of-day:** $F(2, 93467) = 9.54$, $p < .001$). That is, although the percentage of backspace keystrokes increased with ~1.6% during two hours of prolonged task performance, both in the morning and in the afternoon (**afternoon** *minus* **morning**$_{linear}$: $z = -0.46$, $p = 1.0$), the increase followed a more quadratic function in the afternoon compared to the morning (**afternoon** *minus* **morning**$_{quadratic}$: $z = -3.95$, $p < 0.001$; see Table 2 and Fig 1B).

## Day-of-week

In addition to the effects of time-on-task and time-of-day, we also looked into changes in typewriting patterns over the workweek. First, we hypothesized that typing performance would decline over the workweek, reflected in an increase in the interkey interval and the percentage of backspaces. Contrary to our expectations, we observed an increase in the interkey interval from 295ms on Monday to 301ms on both Tuesday (Mon-Tue: $z = -3.93$, $p < .001$) and Wednesday (Mon-Wed: $z = -3.99$, $p < .001$), after which the interkey interval decreased to

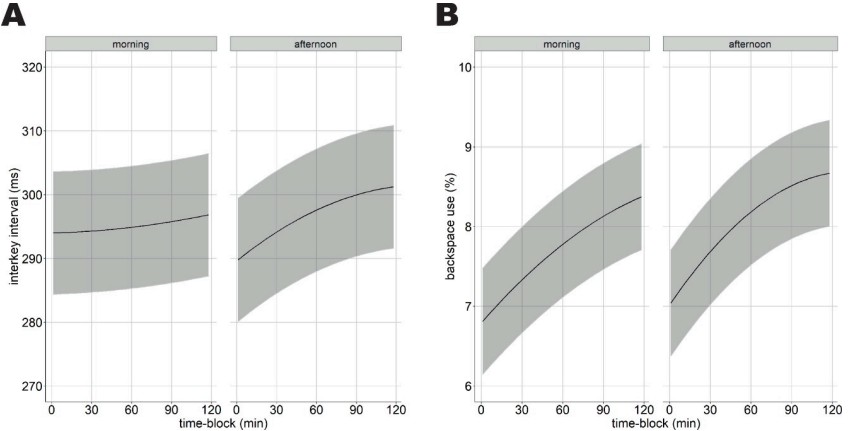

**Fig 1. The effect of time-on-task on typewriting changes from the morning to the afternoon.** Time-blocks are calculated based on the preceding 15 min, see method section. (A) the interaction between time-on-task and time-of-day on the average interkey interval. (B) The interaction between time-on-task and time-of-day on backspace use. The confidence intervals reflect the standard errors of the mean.

**Table 5. Average typing performance on the different days of the week, reflected by the interkey interval (ms) and backspace use (% of backspace keystrokes).**

| Day-of-week | Mean interkey interval in ms (SE) | Mean percentage of backspace keystrokes (SE) |
|---|---|---|
| Monday | 295 (9.62) | 8.62 (0.67) |
| Tuesday | 301 (9.60) | 8.40 (0.66) |
| Wednesday | 301 (9.60) | 7.82 (0.66) |
| Thursday | 296 (9.63) | 7.83 (0.66) |
| Friday | 291 (9.65) | 8.00 (0.67) |

296ms on Thursday and 291ms on Friday ($z_{linear}$ = -6.09, $p$ < .001), during which employees' typewriting was fastest (see Table 5 and Fig 2A).

Backspace use was highest on Monday with 8.6% of the keystrokes were backspace keystrokes, followed by Tuesday with 8.4% compared to the other days of the week (Mon-mean (Wed, Thu, Fri): $z$ = 11.39, $p$ < .001; Tue-mean(Wed, Thu, Fri): $z$ = 8.91, $p$ < .001; see Table 5 and Fig 2B). Backspace use did not significantly differ between Wednesday, Thursday and Friday (Wed-Thu: $z$ = -0.11, $p$ = 1.0; Wed-Fri: $z$ = 2.11, $p$ = .351; Thu-Fri: $z$ = 1.88, $p$ = .604).

Additionally, we observed that the effects of time-on-task on typing speed differed over the days of the week (see Table 2 for an overview of the main and interaction effects). That is, on Monday afternoon ($z$ = -7.10, $p$ < .001), and Friday morning ($z$ = -8.33, $p$ < .001) the interkey interval decreased with time-on-task. On the other days of the week, the interkey interval either remained stable or increased, the last one reflecting a decrease in typing speed with time-on-task.

We also observed changes in the effect of prolonged task performance on backspace use over the working week (see Table 3 for an overview of the main and interaction effects). On all days, except for Friday afternoon, the percentage of backspace keystrokes increased with time-on-task in the morning and in the afternoon. On Friday in the afternoon no change in backspace use was observed ($z$ = 0.82, *n.s.*).

## Speed and accuracy

In order to investigate the relationship between typing speed (interkey interval) and typing accuracy (backspace use) between the morning and the afternoon, we calculated the effect of

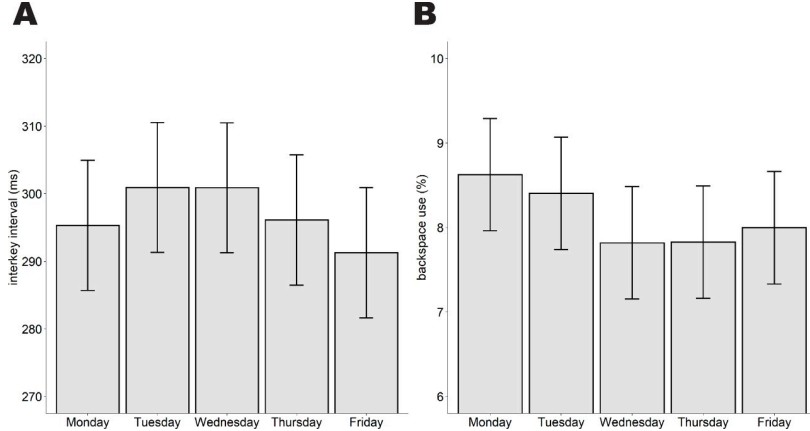

**Fig 2. The course of typewriting performance over the different days of the week.** (A) The effect of day-of-week on the interkey interval. (B) The effect of day-of-week on the percentage of backspace keystrokes. The confidence intervals reflect the standard errors of the mean.

time-on-task in the morning and in the afternoon. The correlations between these coefficients revealed that the relation between speed and accuracy differed between the morning and the afternoon. In the morning, we observed no correlation between the effect of time-on-task on typing speed and the effect of time-on-task on the percentage of backspace keystrokes ($r = -0.04$, *n.s.*). In the afternoon, however, there was a positive relationship between the increase in the interkey interval and the increase in the percentage of backspaces. More specifically, participants that showed a larger increase in the interkey interval with time-on-task also showed a larger increase in the percentage of backspace keystrokes with time-on-task ($r = 0.622$, $p < .001$), indicating that changes in typewriting with time-on-task do not reflect changes in speed-accuracy trade-off.

## Discussion

In the current study we evaluated novel, non-invasive measures based on typewriting to continuously monitor behavior in a working environment. Our aims were, first, to provide a proof of principle of this method, and, second, to study how typewriting dynamics during regular office work change over different time-scales (i.e., time-on-task, time-of-day, day-of-week). Based on earlier findings observed in a controlled environment, we focused on interkey interval and backspace use as indices of behavior. To investigate these aims, the typewriting markers were recorded for six consecutive weeks during regular office work performed in a university environment. We confirmed that typewriting behavior contains sensitive markers that reflect changes in behavior over time. In addition to general changes in speed and accuracy with time-on-task, we found that the effects of time-on-task as indexed by our typewriting measures changed throughout the day. More specifically, the effect of time-on-task on typing speed (i.e., interkey interval) was more pronounced in the afternoon than in the morning. Moreover, on average, office workers used the backspace key more often in the afternoon compared to the morning, although the effect of time-on-task on backspace use, reflecting task accuracy, was smaller in the afternoon. Finally, an analysis of time of week effects provides no evidence for a general decline in performance over the week.

With regard to our first aim, as hypothesized, the length of the interkey interval and the percentage of backspace keystrokes both increased with time-on-task, replicating previous work suggesting that changes in markers derived from typewriting are sensitive to mental fatigue elicited during continuous task performance [18, 36]. Previously, this type of research was mainly conducted in simulated office environments [37], or focused on self-reported behavior of employees [38], using measures that either interrupted regular office work or relied on subjective measures influenced by the observer's personal judgment [39]. Our findings show that our measures based on typing behavior have practical potential to objectively monitor performance efficiency without disturbing regular work-related activities.

A similar pattern of results was observed during simulated office work [18] as in the present, real-life environment. Moreover, the changes with time-on-task in typewriting performance were even found to be more pronounced in the present study. Relevant in this perspective is that in the present study, compared to a relatively controlled experimental environment, many uncontrollable factors may have influenced performance efficiency due to the dynamic nature of the actual office environment [40, 41]. On the one hand, factors such as interruptions related to the presence of others [42] and uncontrollable requests for actions from electronic devices (e.g., online activity, telephone calls), might increase task demands [43], which could in turn increase the effects of mental fatigue on performance efficiency. However, on the other hand, work motivation [1] and enhanced autonomy with regard to setting one's own schedule and planning work-breaks if needed [37], might reduce experienced

task demands and related levels of mental fatigue during regular office work compared to the lab setting. Interestingly, despite these noisy conditions, we observed significant changes in the typewriting indices during prolonged task performance. To summarize, with regard to our first aim, we provided a proof of principle of the sensitivity of these measures, confirming that typewriting markers are susceptible to changes in behavior related to the effects of mental fatigue, not only in a controlled experimental setting, but also in an uncontrolled office environment.

Under real life conditions, factors that influence our behavior vary every day and even from hour to hour, and therefore a substantial variability in performance and the effects of mental fatigue might be expected over time. Our second aim was to investigate how typewriting dynamics during regular office work changed over different time-scales. In general, performance efficiency during real life activities, such as typewriting, depends on two dimensions: speed and accuracy. In the present study, the interkey interval served as an indicator of typing speed and the percentage of backspace keystrokes was used as an indicator of typing accuracy [18]. Using the backspace key is an indirect measure of typing accuracy, given that it is used to correct mistakes in typewriting. Therefore, while interpreting the results, it is important to keep in mind that an increase in backspace use could originate from different types of behavior. That is, in our study, participants could have corrected more (in)correctly typed letters and/or detected their errors later, which, as a result, required more consecutive backspaces to correct one incorrect keystroke.

The results of the present study showed that, in the morning, typing speed remained relatively stable over time. Simultaneously, typing accuracy declined, which was revealed by an increase in backspace use. In the afternoon, we observed a decline in both dimensions of typing performance. More specifically, typing speed decreased over time, reflected by an increase in the interkey interval, and additionally the quality of typing was reduced, indicated by the increase in the percentage of backspace keystrokes with prolonged task performance. This pattern shows similarities with previous research investigating the effects of mental fatigue on task control in a lab setting [44]. Lorist and colleagues showed that if participants were instructed to perform fast, accuracy steadily declined from the start of the experiment, while participants kept responding at a stable speed. After a while, participants seemed to adjust their strategy. That is, over time, participants performed at a slower pace as well, which was observed in an increase in RTs.

In daily life, people adoptively invoke qualitatively different performance strategies. Adopting a strategy that focuses on speed generally results in a larger number of errors, while adopting a strategy that focuses on high accuracy results in slower performance [45]. People tend to moderate this speed-accuracy trade-off based on external conditions and the time available to complete their work. The results suggested that, in the present study, office workers first tried to maintain typing speed, which resulted in an increased number of typing errors they had to correct. In the afternoon, however, they seemed to readjust their strategy, which resulted in a decline in both dimensions of performance. In comparison to the study of Lorist and colleagues [44], who measured prolonged performance during a 2h session, the pattern we found stretched out over the day, indicating that mental fatigue might build up during a working day. These findings imply that, although office workers are entitled to have breaks during a working day, the scheduled breaks might not have been enough to fully recover from the demands they encountered during the day [46].

Previous research on the effects of mental fatigue on typewriting strategies showed that people tend to make more typing errors during prolonged task performance [18]. However, when given the option to correct their mistakes, they at least partly correct these mistakes. Although several studies observed a similar increase in correction behavior during prolonged task

performance, people are not able to correct for the total increase in mistakes, even if these mistakes are in plain sight, that is, clearly visible on the screen in front of them [1, 2]. In order to safeguard the privacy of the employees, however, we did not monitor the identity of the keys apart from the backspace keys. For this reason, we were not able to identify and analyze the mistakes that were not corrected during typewriting.

Error-corrections indirectly reflect accuracy on a given task, and using error-corrections during typewriting as a measure of mental fatigue might therefore provide information on the effects of mental fatigue on underlying cognitive processes. Monitoring performance requires higher-order mental functions, which are prone to the effects of mental fatigue. Experimental studies showed that erroneous responses are usually followed by specific brain activation patterns, called the error-related negativity (ERN) [47, 48], and result in decreased response speed on the next trial (i.e. post-error slowing) [49]. Lorist and colleagues [1] investigated these behavioral and brain activity patterns during prolonged performance on an Eriksen flanker task. They found that performance monitoring declined over time, which was reflected in a significant decrease of brain activity patterns related to error processing (ERN), and was accompanied by a decrease of post-error slowing. This decreased ability to monitor behavior and adapt performance concurrently might have resulted in a later detection of errors and therefore in an increase in error-corrections during the present study, as was also shown in the controlled lab study of de Jong and colleagues [18].

The present study focused on the dynamics in typewriting during prolonged task performance. Previous research repeatedly showed that, in addition to a tonic decline in speed and accuracy over time, fatigued participants also experience short-term lapses in performance during which they are unable to process any information [50, 51]. These phasic lapses in performance, so-called mental blocks, are characterized by extremely long reaction times during experimental tasks and can be detected by studying the distribution of reaction times [25]. In the present study, we excluded lapses of attention by excluding interkey intervals that were longer than 5 seconds. However, for future research it would be interesting to investigate whether the effects of prolonged task performance on length or number of short-term lapses in performance during prolonged typewriting follow a similar pattern as the tonic effects of prolonged task performance on typewriting.

Besides the effects of mental fatigue on typewriting dynamics during the day, the present study also provides direct insight into typewriting dynamics during a working week. First, we found no evidence for a general decline in typewriting performance with day-of-week, given that backspace use remains stable on Wednesday, Thursday and Friday, and the interkey interval decreases, reflecting a increase in typing speed, from Wednesday to Friday. Second, we found that the effects of prolonged task performance on typing speed and accuracy followed a similar pattern over the different days of the week, suggesting that mental fatigue elicited on the previous day, as reflected in the effects of time-on-task and time-of-day, did not influence the course of performance during prolonged task performance on the next day. These results provide proof that mental fatigue does not accumulate across the days of the week. Although previous literature does not paint a consistent picture, Persson and colleagues [52] indicated that alertness of construction workers did not increase during a working week. This pattern was shown in construction workers with a regular working schedule (7–15 h), but also in workers with an extended schedule (six days in a row, one day off, five days in a row, nine days off) that stayed at accommodations at the construction site during the working week.

Our findings provide further evidence for office workers' ability to recover from work-related demands during the week. Nonetheless, typewriting dynamics were subject to daily variations. That is, on Mondays and Fridays, office workers adopted a typewriting strategy that maximized typing speed, while on the other days of the week they either adopted a strategy

that maximized accuracy (Wednesdays), or performed both fast and accurate compared to the other days of the week (Thursdays). Although some studies showed that employees recover from regular work-related demands after engaging in pleasurable activities during the weekend [23, 53], other studies revealed that Mondays serve as a transition day from pleasurable activities to the structured demanding work week, which is reflected in a more negative mood [54], increased stress levels [55], and decreased ability to recover from work-related demands on Mondays [56]. These factors might have also led to the observed behavioral patterns in the present study. Similarly, it could be argued that Fridays also serves as a transition day from the work week to the weekend. In contrast to Mondays, however, Fridays have previously been associated with improved mood compared to the rest of the week [57].

This study has implications for real-life working environments, given that a large part of the working population regularly performs computer work. In the Netherlands, for example, 40% of the employees perform computer work more than 6 h every day [58]. There are several ways in which monitoring typewriting could support employees during their work. First, personalized real-time feedback based on changes in typing behavior could be provided to the users in order to help them detect when lapses in performance occur and a short break might be beneficial. However, real-time feedback might be biased due to dynamics in typewriting performance that are not related to lapses in performance. One of the characteristics of our working environment is the large variability in working conditions, due to changes in work-related tasks, noise in the working environment, and changes in general persons state, among others. Our method also allows monitoring performance over a longer period of time enabling us to detect regularities in working activities. Related to this, a second possibility of our method is to provide feedback on an individual level to help employees realize a more optimal work-break schedule that is complementary with their individual state and specific work-related demands. By comparing behavior dynamics over several weeks, typing behavior could help decide when, during the workday or -week an employee should work on tasks that need high accuracy or when it is better to work on less demanding tasks. A third option is to use changes in typing behavior to evaluate interventions in the working environment. For instance, it might provide relevant information with regard to performance efficiency for evaluating the effectiveness of a 6-hour workday instead of our regular 8-hour workday. Previously, researchers already used questionnaires to evaluate this specific intervention, however, measuring performance, and importantly, doing so without interrupting regular activities, could enhance our knowledge of its effects on performance and productivity more objectively.

## Conclusions

The typing indices that were used to describe behavior dynamics reflect subtitle changes in both speed and accuracy during regular office work, not only during the day but also over the week. These findings might be relevant to consider when scheduling different tasks over the day, but could also provide information about the number of hours that employees can or should work during a day.

## Supporting information

**S1 Appendix.**
(DOCX)

**S2 Appendix.**
(DOCX)

## Acknowledgments

The authors would like to thank Aafke Wiekens for recruiting the participants.

## Author Contributions

**Conceptualization:** Marlon de Jong, Jacob Jolij, Monicque M. Lorist.

**Data curation:** Marlon de Jong, Anne M. Bonvanie.

**Formal analysis:** Marlon de Jong, Monicque M. Lorist.

**Funding acquisition:** Monicque M. Lorist.

**Investigation:** Marlon de Jong.

**Methodology:** Marlon de Jong, Anne M. Bonvanie, Monicque M. Lorist.

**Resources:** Anne M. Bonvanie.

**Software:** Marlon de Jong, Anne M. Bonvanie.

**Supervision:** Jacob Jolij, Monicque M. Lorist.

**Visualization:** Marlon de Jong.

**Writing – original draft:** Marlon de Jong, Anne M. Bonvanie, Jacob Jolij, Monicque M. Lorist.

**Writing – review & editing:** Marlon de Jong, Monicque M. Lorist.

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
