## [Decision Letter · Decision Letter 0]

11 May 2020

PONE-D-20-10944

Dynamics in typewriting performance reflect mental fatigue during real-life office work

PLOS ONE

Dear Dr. Lorist,

Thank you for submitting your manuscript to PLOS ONE. Two experts commented on your manuscript. As you can see from the reviews, both referees found the general topic addressed in your manuscript interesting and they have a number of nice things to say about the study. At the same time, they have some remarkably constructive and excellently detailed suggestions how to further improve the paper. The comments speak for themselves, but it is obvious that one reoccurring theme is the need for more specificity regarding the hypotheses and a more systematic presentation of the statistics/results. While this will require some extra efforts, I consider it worthwhile. Hence, we invite you to submit a revision of the manuscript that addresses the remaining points together with a cover letter that contains point-by-point replies. Some additional editorial comments are added below. 

 Best regards,Michael B. Steinborn, PhDAcademic Editor

We would appreciate receiving your revised manuscript by Jun 25 2020 11:59PM. To enhance the reproducibility of your results, we recommend that if applicable you deposit your laboratory protocols in protocols.io, where a protocol can be assigned its own identifier (DOI) such that it can be cited independently in the future. For instructions see: http://journals.plos.org/plosone/s/submission-guidelines#loc-laboratory-protocols

We look forward to receiving your revised manuscript.

Kind regards,

Michael B. Steinborn, PhD

Academic Editor

PLOS ONE

Journal Requirements:

Additional Editor Comments:

 line 111-220methods/statistics: statistical terms (M, SD, F, p, etc.) should be written in italics  line 251-262a distinctive feature is that the typing task requires self-paced responding (using the inter-key interval to index reaction time). Under these conditions, individuals are highly vulnerable to occasional lapses (or blocks) of performance and this tendency might also increase with time on task. This has important theoretical consequences as the time on task effect is not due to a tonic slowdown of performance speed but originates from an increase in the number of these short-term depletion (or lapses) in performance (which add to the mean performance). Individuals during a mental blockade might be entirely unable to process any information until the mental blockade dissipates, which can only be demonstrated with advanced performance measurement methods, using distributional analysis. My own work is relevant with this regard (Steinborn & Huestegge, 2016) as we have set a methodological benchmark of how to measure these effects accurately in performance settings [Steinborn, M. B., & Huestegge, L. (2016). A walk down the lane gives wings to your brain: Restorative benefits of rest  breaks on cognition and self-control. Applied Cognitive Psychology, 30(5), 795-805. doi:10.1002/acp.3255]. While I would not demand further analysis of typical speed variability (i would welcome it, however), I would appreciate if the authors could address this point in the revised version of the manuscript.   line 273-285backspace key to index erroneous behavior. In my opinion, this is a highly interesting point that should be given somewhat more weight in the discussion. As most of the research is using reaction-time based paradigms where it is not possible to correct an error, the present study uses a natural setting where an error is completely relevant to the task and where correction is not only enabled but naturally invited by the nature of the task. I suggest elaborating this point a bit further, if possible, as it would increase the impact of the present study.    

Reviewers' comments:

Reviewer's Responses to Questions

**Comments to the Author**

1. Is the manuscript technically sound, and do the data support the conclusions?

Reviewer #1: Yes

Reviewer #2: Yes

2. Has the statistical analysis been performed appropriately and rigorously? 

Reviewer #1: Yes

Reviewer #2: Yes

3. Have the authors made all data underlying the findings in their manuscript fully available?

Reviewer #1: Yes

Reviewer #2: No

4. Is the manuscript presented in an intelligible fashion and written in standard English?

Reviewer #1: Yes

Reviewer #2: Yes

5. Review Comments to the Author

Reviewer #1: This paper examines the effect of time-of-week, time-of-day, and time-on-task on type writing performance in a natural setting, using a sample of about 50 participants. Type writing performance was measured using typing speed (inter key interval) and error correction (backspace) as dependent measures. It was hypothesised that the development of task-related mental fatigue (time on task) is modulated by time of week (early weekdays better than late weekday) and time of day (afternoon times better than morning times). It seems that the results support the hypothesis although some aspects of the results are not completely clear to me at the moment. My evaluation is positive, and I have a few comments that might be considered in the revision.

#1 Theory

It is clear to me that the authors are the "top experts" in that field of research on mental fatigue. I am therefore very much inclined to believe most of what is stated in the introduction. Apart from that, there are some points that could be explained in more detail or with more precision. For example, the underlying processes that are assumed to produce the performance variations at different time scale, over the week or with time on task, could be specified with a bit more precision. In other words, the rationale for expecting interactions between these variables on the relevant performance measures could be explained further. On the other hand, the manuscript is relatively concise at present and the might be little room to include too much detail, so I only ask (as an interested reader) whether the authors could elaborate a bit more on the underlying cognitive mechanisms of mental fatigue in the introduction and also in the discussion.

#2 Hypotheses / Statistics

The hypotheses are so far well formulated, however, they could be more systematically presented with respect to all relevant main effects and interactive effects on performance. Regarding the statistics, I suggest including a table that contains all main and interaction effects on performance. At the moment, I did not fully understand all aspects of the results and I therefore would appreciate if this information is more systematically presented in the revised manuscript.

#3 Results

Although the authors are interested in some more specific effects, I would appreciate if they could provide a full model of all main and interaction effects on both performance measures (typing speed, and error correction). For example, there are expected main effects of the factor "time of week", of the factor time of day, and also on "time on task", then three two-way interactions and one three-way interaction, resulting in seven relevant statistical effects. A more integrative presentation of these results would certainly improve the current manuscript.

#4 self-report state

Performance is evidently influenced by momentary states within the individual. Given the evidence of relationships between subjective engagement to a task and objective performance, I wonder whether self-report measures are available for the present research, and if so, whether they could deliver additional information. For example, the dundee stress state questionnaire (DSSQ, Matthews et al., 2002; Langner et al., 2010) seems to be the proper instrument to assess these aspects but other instruments might also do well with this regard. I would appreciate if the authors could give a short opinion or outlook on the possibilities of assessing engagement and to elaborate somewhat more deeply on potential limitations with this regard in the present study [top references: [Langner, R. et al. (2010). Mental fatigue and temporal preparation in simple reaction-time performance. Acta Psychologica, 133(1), 64-72. doi:10.1016/j.actpsy.2009.10.001; Matthews, G. et al. (2002). Fundamental dimensions of subjective state in performance settings: Task engagement, distress, and worry. Emotion, 2(4), 315-340. doi:10.1037//1528-3542.2.4.315].

#5 minors and typoes

line 165-168, table 1

table according to APA rules, provide sufficient information as notes

line 170-end of results

F values should be rounded up (2,97924 to 3.0)

line 185-190, figure 1

if possible, the results should be presented not separately but in one figure so that the reader can evaluate the main and interactive effects of the results simultaneously. At present, the results are distributed across separated figures which makes it difficult to understand the whole picture

line 349, references

check typos

Reviewer #2: Background:

The study examines how mental fatigue, measured by markers of typewriting performance for speed and accuracy, changes over different time scales during regular office work.

To this end, the authors aimed to:

1. provide a proof-of-principle regarding two formerly identified markers of typewriting performance

2. investigate changes in typewriting performance over different time scales

As a result, differences in typewriting performance were found for the different time scales under investigation. These differences resemble those found in an earlier study, where a direct link between typewriting performance and mental fatigue was established.

Evaluation:

Overall, my evaluation is positive, especially if a few points are further clarified. Except for a few minor exceptions, the manuscript is a pleasurable read. The document is formally well written, mostly to the point and adequate in length. A few less strong points regard the elaboration of the effective mechanism under investigation and the statistical reporting. I believe there is potential for improvement with this regard. My detailed comments are outlined below. Please note that my comments are aimed at further improving the manuscript, and are not meant to criticize the authors' work.

1. Theory

The predictions would benefit from further elaboration. While it is understood that the proof-of-principle character of the study somewhat mitigates this point, it is so far not clear what exactly is predicted for either of the different time-scales. Thus, the reported results for performance for either time-scale might point to an effect of mental fatigue, they might, however, also be somewhat unrelated to mental fatigue and caused by additional factors. I would therefore expect the authors to provide further clarity regarding their expectations for each outcome measure and time-scale.

2. Statistical reporting

The statistical reporting part lacks a systematic and comprehensive overview of the reported outcome measures. Given this lack, it is thus far not possible to readily understand either the main effects or the interactions found for typewriting performance. Statistics should thus be systematically described in tables, reporting all main and interaction effects. A detailed table of complete results as well as some sort of reporting of the compiled data (means, standard deviations, ... for all days and time-scales seperately as well as combined) would also greatly improve the readers' ability to comprehend the reported results.

3. Methods

- potential trade-off between speed and accuracy

As it stands, typewriting markers for speed and accuracy are not measured independently of each other and might thus be confounded. A faster typing speed might result in more necessary backstrokes to correct a mistake. Similarly, more mistakes and thus more (potentially rather fast) backstrokes might lead to an enhanced typing speed. Given the design of the study, both these effects would not be correctly reflected by the current measures. While I understand that an independent measurement might be difficult to achieve, I suggest the authors should provide some further analysis of how this problem might affect the found results.

- sample bias

The comparatively large drop-out ratio appears problematic in regards to a potential sample bias, especially given the rather strict criteria of more than 45 minutes of uninterrupted work for at least 30 times a week. Several potential solutions come to mind and are highly recommened to provide further clarity regarding this point:

- reconsider the strict criteria or give a more comprehensive explanation for its choice

- discuss potential effects from excluding a large portion of the sample, especially regarding the potential that mental fatigue might might be more pronounced in the dropped-out participants (thus leading to a smaller amount of unterinterrupted work)

Line comments:

line 107 (theory): "... expected that typing performance would ..." This hypothesis would especially benefit from further clarification, particularly since no overall measure of performance is given and therefore no possible way to judge whether hypothesis will be approved or rejected if speed and accuracy fail to change in the expected dimension

line 112-120 (methods - participants): please provide a more detailed description of the work the participants carry out and on what exactly is typed by them (predominantly email, research articles, ...)

line 118 (methods - participants): please clarifiy what is meant by continous typing

line 134-136 (methods - typing performance): it is not easily understood how the series of average values are generated, please provide some more details on this

line 146-147 (methods - procedure): please explain the nature of the feedback in more detail. It might also be beneficial to discuss the implications of the given feedback on performance and the given results, if any effect is expected

line 205 (results - day-of-week): given that most people work less hours on Fridays, performance on Friday afternoons might lack data points. I suggest to provide more comprehensive and complete results to understand how potential effects like these were treated.

line 311 (discussion): Given that mental fatigue was measured only on weekdays, but might also occur on Saturdays and Sundays, such a general conclusion might be exagerated to a certain degree. Thus, "baseline" measurement on Mondays might not reflect a true baseline for mental fatigue.

line 329 (discussion): typo "lead", should read "led"

line 333 (discussion): typo "subtitle", should read "subtle"

line 336-337 (discussion): please explain in greater detail how the presented results point to information about the amount of hours employees can or should work

Figure 2: please note what is indicated by confidence intervals

6. PLOS authors have the option to publish the peer review history of their article (what does this mean?). If published, this will include your full peer review and any attached files.

Reviewer #1: No

Reviewer #2: No

---

## [Author Response · Author response to Decision Letter 0]

24 Aug 2020

Response to reviewer and editor comments

Journal Requirements:

 The style requirements were checked and applied.

We included two questionnaires in S1 Appendix and S2 Appendix. The questionnaire in S1 Appendix includes demographic and work-related questions. The weekly questionnaire which is included in S2 Appendix focused on how participants experienced the week before. 

There are legal restrictions on sharing the data:

“The data cannot be publicly shared because the data is considered to be personal data under the GDPR, and cannot be further anonymized. The research participants have not given their consent for sharing data beyond the original research purpose, which does include data access for independent verification of the results, but prohibits public sharing. These limitations are imposed by European law and verified by the data office of the faculty of Behavioral and Social Sciences.

Data access can be requested via the data office of the faculty of Behavioral and Social Sciences of the University of Groningen: research-data-bss@rug.nl. Please quote study code: #2017-03_02 ECFEB in the request.”

Additional Editor Comments:

line 111-220

methods/statistics: statistical terms (M, SD, F, p, etc.) should be written in italics

We would like to thank the editor for pointing this out. We changed this throughout the manuscript. 

line 251-262

a distinctive feature is that the typing task requires self-paced responding (using the inter-key interval to index reaction time). Under these conditions, individuals are highly vulnerable to occasional lapses (or blocks) of performance and this tendency might also increase with time on task. This has important theoretical consequences as the time on task effect is not due to a tonic slowdown of performance speed but originates from an increase in the number of these short-term depletion (or lapses) in performance (which add to the mean performance). Individuals during a mental blockade might be entirely unable to process any information until the mental blockade dissipates, which can only be demonstrated with advanced performance measurement methods, using distributional analysis. My own work is relevant with this regard (Steinborn & Huestegge, 2016) as we have set a methodological benchmark of how to measure these effects accurately in performance settings [Steinborn, M. B., & Huestegge, L. (2016). A walk down the lane gives wings to your brain: Restorative benefits of rest breaks on cognition and self-control. Applied Cognitive Psychology, 30(5), 795-805. doi:10.1002/acp.3255]. While I would not demand further analysis of typical speed variability (i would welcome it, however), I would appreciate if the authors could address this point in the revised version of the manuscript. 

We enjoyed reading the article and agree that the manuscript would benefit from discussing these findings. In the present manuscript, we analyzed average interkey intervals and percentage of backspaces, therefore it was not possible to compare the distribution of the individual responses, as was described in the study of Steinborn and Huestegge (2016). However, we looked into the data of our previous study (de Jong et al., 2018), where we investigated the effects of prolonged task performance on typewriting in an experimental setting. Here, we did neither observe a change in the distribution of the interkey intervals over time, nor an increase in extremely large interkey intervals. Still, we agree that it would be interesting to highlight this part of the literature and encourage further research into lapses of attention in the working environment. Therefore, we added a paragraph in the discussion on page 23, lines 407-420: 

“The present study focused on the dynamics in typewriting during prolonged task performance. Previous research repeatedly showed that, in addition to a tonic decline in speed and accuracy over time, fatigued participants also experience short-term lapses in performance during which they are unable to process any information (Bills, 1931; Bertelson & Joffe, 1963). These phasic lapses in performance, so-called mental blocks, are characterized by extremely long reaction times during experimental tasks and can be detected by studying the distribution of reaction times (Steinborn and Huestegge, 2016). In the present study, we excluded lapses of attention by excluding interkey intervals that were longer than 5 seconds. However, for future research it would be interesting to investigate whether the effects of prolonged task performance on length or number of short-term lapses in performance during prolonged typewriting follow a similar pattern as the tonic effects of prolonged task performance on typewriting.”

line 273-285

backspace key to index erroneous behavior. In my opinion, this is a highly interesting point that should be given somewhat more weight in the discussion. As most of the research is using reaction-time based paradigms where it is not possible to correct an error, the present study uses a natural setting where an error is completely relevant to the task and where correction is not only enabled but naturally invited by the nature of the task. I suggest elaborating this point a bit further, if possible, as it would increase the impact of the present study. 

We agree with the editor that using correction behavior is an interesting index of erroneous behavior, especially in mental fatigue research. After re-reading the manuscript, we also agree that this measure deserves a more elaborate discussion. In the discussion of the manuscript we elaborated this point somewhat further in the discussion on page 23, lines 421-431:

“Error-corrections directly reflect accuracy on a given task, and using error-corrections during typewriting as a measure of mental fatigue might therefore provide information on the effects of mental fatigue on underlying cognitive processes. Monitoring performance requires higher-order mental functions, which are prone to the effects of mental fatigue. Experimental studies showed that erroneous responses are usually followed by specific brain activation patterns, called the error-related negativity (ERN; Falkenstein et al., 1991; Gehring et al., 1993), and result in decreased response speed on the next trial (i.e. post-error slowing; e.g., Rabbit, 1966). Lorist and colleagues (2005) investigated these behavioral and brain activity patterns during prolonged performance on an Eriksen flanker task. They found that performance monitoring declined over time, which was reflected in a significant decrease of brain activity patterns related to error processing (ERN), and was accompanied by a decrease of post-error slowing. This decreased ability to monitor behavior and adapt performance concurrently might have resulted in a later detection of errors and therefore in an increase in error-corrections during the present study, as was also shown in the controlled lab study of de Jong and colleagues (2018).”

In addition, we added a sentence to the abstract (page 2, lines 29-30) to make the relevance of backspace use as an indicator of erroneous behavior in a natural environment more clear to the reader.

“allowing not only to examine performance speed, but also providing a natural setting to study error correction.”

Comments to the Author

Reviewer #1: 

This paper examines the effect of time-of-week, time-of-day, and time-on-task on type writing performance in a natural setting, using a sample of about 50 participants. Type writing performance was measured using typing speed (inter key interval) and error correction (backspace) as dependent measures. It was hypothesised that the development of task-related mental fatigue (time on task) is modulated by time of week (early weekdays better than late weekday) and time of day (afternoon times better than morning times). It seems that the results support the hypothesis although some aspects of the results are not completely clear to me at the moment. My evaluation is positive, and I have a few comments that might be considered in the revision.

We would like to thank the reviewer for their thoughtful comments. As a result we were able to greatly improve the manuscript.

 #1 Theory

It is clear to me that the authors are the "top experts" in that field of research on mental fatigue. I am therefore very much inclined to believe most of what is stated in the introduction. Apart from that, there are some points that could be explained in more detail or with more precision. For example, the underlying processes that are assumed to produce the performance variations at different time scale, over the week or with time on task, could be specified with a bit more precision. In other words, the rationale for expecting interactions between these variables on the relevant performance measures could be explained further. On the other hand, the manuscript is relatively concise at present and the might be little room to include too much detail, so I only ask (as an interested reader) whether the authors could elaborate a bit more on the underlying cognitive mechanisms of mental fatigue in the introduction and also in the discussion.

Based on these suggestions, we changed the manuscript in three ways, which are discussed below:

#1 We agree with the reviewer that it would benefit the manuscript if we elaborate a bit more on the underlying cognitive mechanisms of mental fatigue. We added a couple of sentences to the introduction on page 3, lines 58-64:

“Several theories regarding the cognitive mechanisms behind its manifestation have been proposed, eventually leading to the widely accepted theory suggesting that mental fatigue develops as a result of a cost-benefit evaluation of effort (Boksem and Tops, 2008; Hockey, 2011). According to this theory, if the costs of performing a task exceed the benefits of finishing the task, people will come to experience subjective feelings of mental fatigue (e.g., aversion against task performance, low vigilance) and performance deteriorates (i.e., people become slower and less accurate).”

#2 In line with the comments of reviewer 2, we also more explicitly discussed the previous research on the effects of prolonged task performance on different time-scales on page 5, lines 107-118. 

“Although there is substantial evidence that prolonged task performance, manipulated by time-on-task, time-of-day, and day-of-week, separately influence performance, interestingly, it is not yet known how these factors interact and subsequently influence behavior dynamics.

Previous experimental studies on mental fatigue mainly focused on the effects of time-on-task on behavioral performance, investigating isolated effects of prolonged task performance on specific cognitive processes (e.g., error processing; Lorist et al., 2005). In addition, studies in real-life settings focused on specific professions, especially those involving shift-work (Brown et al., 2020; Kecklund & Axelsson, 2016), where the manifestation of fatigue was expected to be potent or even dangerous, given its relationship with serious accidents (Akerstedt & Haraldsson, 2001; Swaen et al., 2002; Chan, 2011). There seems to have been done little research on the manifestation of mental fatigue during regular 9 to 5 jobs.”

#3 The editor made a comment about more explicitly discussing the use of correction behavior as an index of erroneous behavior, especially in mental fatigue research. We took this opportunity to more elaborately discuss the cognitive processes underlying erroneous behavior, which have been found to be affected by mental fatigue. See our reaction to this comment of the editor. 

#2 Hypotheses / Statistics

The hypotheses are so far well formulated, however, they could be more systematically presented with respect to all relevant main effects and interactive effects on performance. Regarding the statistics, I suggest including a table that contains all main and interaction effects on performance. At the moment, I did not fully understand all aspects of the results and I therefore would appreciate if this information is more systematically presented in the revised manuscript.

We would like to thank the reviewer for pointing this out. Both reviewer 1 and 2 asked for clarification of the hypotheses and the statistics. Based on these comments, we more systematically presented the hypotheses in the introduction on page 6, lines 128-138: 

“In line with findings in an experimental setting (de Jong et al., 2018), we hypothesized that there would be a main effect of time-on-task on both the interkey interval and the percentage of backspaces, where we expected that both measures would increase with time-on-task. Secondly, we hypothesized that the magnitude of the effect of time-on-task on these performance measures would depend on time-of-day (main effect). That is, we expected a larger increase in both the interkey interval and the percentage of backspaces with time-on-task in the afternoon (interaction) than in the morning. Lastly, we hypothesized that typewriting patterns would change over the course of the week. We expected these changes to manifest in two ways. First, we hypothesized that employees became slower and less accurate over the week (main effect), and second, we expected a larger decline of performance (interkey interval and backspace use) with time-on-task over the week (interaction).”

Additionally, we re-analyzed the data and rewrote the result section, including several tables that reflect the main and interaction effects. These were included to address the results more systematically. We also changed the design by including the interaction of time-on-task and time-of-day with day-of-week. Although this affected some of the results, these revisions did not change our main conclusions. 

#3 Results

Although the authors are interested in some more specific effects, I would appreciate if they could provide a full model of all main and interaction effects on both performance measures (typing speed, and error correction). For example, there are expected main effects of the factor "time of week", of the factor time of day, and also on "time on task", then three two-way interactions and one three-way interaction, resulting in seven relevant statistical effects. A more integrative presentation of these results would certainly improve the current manuscript.

As we already mentioned in our response to the previous comment, we completely rewrote the result section including multiple tables to address the results more systematically. We hope our results are now more clear.

#4 self-report state

Performance is evidently influenced by momentary states within the individual. Given the evidence of relationships between subjective engagement to a task and objective performance, I wonder whether self-report measures are available for the present research, and if so, whether they could deliver additional information. For example, the dundee stress state questionnaire (DSSQ, Matthews et al., 2002; Langner et al., 2010) seems to be the proper instrument to assess these aspects but other instruments might also do well with this regard. I would appreciate if the authors could give a short opinion or outlook on the possibilities of assessing engagement and to elaborate somewhat more deeply on potential limitations with this regard in the present study [top references: [Langner, R. et al. (2010). Mental fatigue and temporal preparation in simple reaction-time performance. Acta Psychologica, 133(1), 64-72. doi:10.1016/j.actpsy.2009.10.001; Matthews, G. et al. (2002). Fundamental dimensions of subjective state in performance settings: Task engagement, distress, and worry. Emotion, 2(4), 315-340. doi:10.1037//1528-3542.2.4.315].

We appreciate that the reviewer would like to know our opinion or outlook on the possibility of assessing subjective engagement. It would definitely have been interesting to measure and compare the participants’ subjective state with the continuous performance measures. In the present study, however, we chose not to measure the participants’ subjective state. The main reason not to include self-report measures is that this study was specifically conducted in order to provide insight into the effects of prolonged task performance during regular working activities without interfering with these activities. Asking participants to fill out questionnaires would have influenced the flow of work. Moreover, since we did not know the exact schedule of the participants beforehand, we neither were able to schedule subjective state measures on a micro level (time-on-task) without interrupting ongoing activities. 

Additionally, based on our previous research on mental fatigue, including our findings in an experimental setting examining typewriting performance (de Jong et al., 2018) showed that subjective fatigue and performance not necessarily change simultaneously. 

Although we did not incorporate subjective engagement questions for any of the time-scales, we agree that adding these questionnaires (e.g., at the end of the different days of the week to prevent interruptions during a working day) might have provided additional insights into the effects we found on Mondays and Fridays, which slightly deviated from the current literature. However, this still would not have enabled us to examine the relationship between the subjective scores and the effects of time-on-task and time-of-day.

#5 minors and typoes

line 165-168, table 1

table according to APA rules, provide sufficient information as notes

We changed the table according to APA rules, and provided additional information in the notes below the table.

line 170-end of results

F values should be rounded up (2, 97924 to 3.0)

F-values need to be reported with degrees of freedom. We noticed that we did not put a space between the comma and the next number. Therefore, it might appear as one number. We changed this throughout the manuscript.

 line 185-190, figure 1

if possible, the results should be presented not separately but in one figure so that the reader can evaluate the main and interactive effects of the results simultaneously. At present, the results are distributed across separated figures which makes it difficult to understand the whole picture

We would like to thank the reviewer for this suggestion. We would like to argue, however, that including all the effects in one figure would not benefit a systematic discussion of our results. We had specific hypotheses that correspond with the included figures. We hope that more systematically discussing these hypotheses and the rewritten result section helps with better understanding our hypotheses and our findings regarding these hypotheses. 

line 349, references

check typos

We now more thoroughly checked the reference list on typos.

Reviewer #2: 

Background:

The study examines how mental fatigue, measured by markers of typewriting performance for speed and accuracy, changes over different time scales during regular office work.

To this end, the authors aimed to:

1. provide a proof-of-principle regarding two formerly identified markers of typewriting performance

2. investigate changes in typewriting performance over different time scales

As a result, differences in typewriting performance were found for the different time scales under investigation. These differences resemble those found in an earlier study, where a direct link between typewriting performance and mental fatigue was established.

Evaluation:

Overall, my evaluation is positive, especially if a few points are further clarified. Except for a few minor exceptions, the manuscript is a pleasurable read. The document is formally well written, mostly to the point and adequate in length. A few less strong points regard the elaboration of the effective mechanism under investigation and the statistical reporting. I believe there is potential for improvement with this regard. My detailed comments are outlined below. Please note that my comments are aimed at further improving the manuscript, and are not meant to criticize the authors' work.

We would like to thank the reviewer for the positive and constructive feedback. Based on the reviewer’s comments we were able to clarify and better present our findings.

 1. Theory

The predictions would benefit from further elaboration. While it is understood that the proof-of-principle character of the study somewhat mitigates this point, it is so far not clear what exactly is predicted for either of the different time-scales. Thus, the reported results for performance for either time-scale might point to an effect of mental fatigue, they might, however, also be somewhat unrelated to mental fatigue and caused by additional factors. I would therefore expect the authors to provide further clarity regarding their expectations for each outcome measure and time-scale.

We would like to thank the reviewer for pointing this out. We agree with the reviewer that the expected changes in performance for the interkey interval and for backspace use were not clearly described. Therefore, we rewrote the hypotheses paragraph on page 6, lines 128-138 (see our response to reviewer 1). 

2. Statistical reporting

 The statistical reporting part lacks a systematic and comprehensive overview of the reported outcome measures. Given this lack, it is thus far not possible to readily understand either the main effects or the interactions found for typewriting performance. Statistics should thus be systematically described in tables, reporting all main and interaction effects. A detailed table of complete results as well as some sort of reporting of the compiled data (means, standard deviations, ... for all days and time-scales seperately as well as combined) would also greatly improve the readers' ability to comprehend the reported results.

We agree with the reviewers and the editor that the statistics lack a systematic and comprehensive overview. Therefore, we rewrote the result section, where we now included multiple tables thereby providing an overview of the interaction and main effects. 

3. Methods

- potential trade-off between speed and accuracy

As it stands, typewriting markers for speed and accuracy are not measured independently of each other and might thus be confounded. A faster typing speed might result in more necessary backstrokes to correct a mistake. Similarly, more mistakes and thus more (potentially rather fast) backstrokes might lead to an enhanced typing speed. Given the design of the study, both these effects would not be correctly reflected by the current measures. While I understand that an independent measurement might be difficult to achieve, I suggest the authors should provide some further analysis of how this problem might affect the found results.

We would like to thank the reviewer for this interesting comment. We compared the effect of time-on-task on the interkey interval and the effect of time-on-task on the percentage of backspace keystrokes. In the morning, we did not find a correlation between these measures, suggesting that a larger interkey intervals with time-on-task was not accompanied by a decrease in backspace keystrokes. In the afternoon, we observed a positive correlation between the effects of time-on-task on both performance measures. That is, participants that showed a larger interkey interval with time-on-task, also used the backspace key more frequently, which means that a larger decline in speed is accompanied by a larger decline in accuracy. 

These results contradict the scenario that is described by the reviewer, and suggest that changes in typewriting with time-on-task do not reflect a change in speed-accuracy trade-off. We rewrote the result section, and reworked the paragraph regarding the relationship between speed and accuracy on page 18 (lines 307-318) in order to make this more clear to the reader. 

 - sample bias

The comparatively large drop-out ratio appears problematic in regards to a potential sample bias, especially given the rather strict criteria of more than 45 minutes of uninterrupted work for at least 30 times a week. Several potential solutions come to mind and are highly recommened to provide further clarity regarding this point:

- reconsider the strict criteria or give a more comprehensive explanation for its choice

- discuss potential effects from excluding a large portion of the sample, especially regarding the potential that mental fatigue might might be more pronounced in the dropped-out participants (thus leading to a smaller amount of unterinterrupted work)

We agree with the reviewer that there is a large drop-out ratio, and that his needs further clarification in our revised manuscript. The participant were employees of the University of Groningen and there was large variation across function profiles (see Table 1) and tasks (e.g., writing research articles, teaching). During this study, several participants were performing teaching activities or were collecting research data, which resulted in limited typewriting, at least in their normal office environment. In addition, part of the participants used multiple work stations (e.g., office computer and laptop), but did not install the keylogging tool on both work stations. Due to these factors, we excluded a large portion of our participants. We clarified this in the method section on page 7, lines 128-138: 

“There was variation in function profile across participants that were included in the study (i.e., scientific staff, support staff). Participants that were excluded from the analyses performed working activities during the measurement period that did not include the required amount of typing activities (e.g., teaching and collecting research data), which was specifically the case for Ph.D. students and (Postdoctoral) researchers. In addition, a number of participants worked on multiple workstations during the 6 weeks of data collection, which was reflected in a limited amount of typewriting data that was recorded from these participants at the workstation on which the recording software was installed. Data of these participants were excluded from the analyses, as well.”

Line comments:

line 107 (theory): "... expected that typing performance would ..." This hypothesis would especially benefit from further clarification, particularly since no overall measure of performance is given and therefore no possible way to judge whether hypothesis will be approved or rejected if speed and accuracy fail to change in the expected dimension

In line with the comment of reviewer 1 and 2 regarding clarification of our predictions, we clarified, and more systematically presented our hypotheses on page 6, lines 128-138. 

line 112-120 (methods - participants): please provide a more detailed description of the work the participants carry out and on what exactly is typed by them (predominantly email, research articles, ...)

We would like to thank the reviewer for pointing this out. Due to privacy concerns, we did not ask the participants about their work activities. 

line 118 (methods - participants): please clarifiy what is meant by continous typing

We clarified what was meant by continuous typing in the method section on page 9, lines 176-178:

“In the present study, continuous typewriting was defined as typewriting during a block of at least 45 minutes.”

line 134-136 (methods - typing performance): it is not easily understood how the series of average values are generated, please provide some more details on this. 

We rewrote the paragraph about typing performance in the method section in order to improve the readers understanding of how the series of average values are generated (page 8, lines 168-178). 

line 146-147 (methods - procedure): please explain the nature of the feedback in more detail. It might also be beneficial to discuss the implications of the given feedback on performance and the given results, if any effect is expected

We included more information on the nature of the feedback on page 9, lines 185-190: 

“During this week, participants also filled out a questionnaire with demographic and work-related questions (S1 Appendix). Each Monday, starting in the second week of the experiment, participants filled out a questionnaire with general questions about how they experienced the week before (S2 Appendix). Each working day, participants received real-time feedback on their performance provided via text messages on their mobile phones and via email. An overview was provided via email at the end of the day.“

Additionally, we analyzed the questionnaire data (5-point likert-scale, where 1 reflects completely agree and 5 reflects completely disagree) in order to investigate to what extend participants were aware of the feedback and whether they used the feedback to adapt their behavior. Although participants read the feedback that was provided to them (M =1.54, SD=0.99), they did not use the feedback to adapt their behavior (M =3.56, SD=1.43). This might be explained by the fact that they did not find the feedback reliable (M=3.20, SD=1.33). 

line 205 (results - day-of-week): given that most people work less hours on Fridays, performance on Friday afternoons might lack data points. I suggest to provide more comprehensive and complete results to understand how potential effects like these were treated.

There were two participants in our sample that did not perform typewriting activities on Fridays and an additional 2 participants that did not perform typewriting activities in the afternoon. Except for baseline differences due to different typewriting style, excluding these participants from the analyses did not influence our results.

line 311 (discussion): Given that mental fatigue was measured only on weekdays, but might also occur on Saturdays and Sundays, such a general conclusion might be exagerated to a certain degree. Thus, "baseline" measurement on Mondays might not reflect a true baseline for mental fatigue.

We agree with the reviewer that measurement on Monday might not have reflected a true baseline. However, backspace use remains stable on Wednesday, Thursday and Friday. Additionally, the interkey interval decreases from Wednesday to Friday, reflecting an increase in typing speed. Moreover, the effects of time-on-task and time-of-day on the interkey interval and backspace use are not more pronounced over the working week. These results suggest that fatigue builds up over the working week. We clarified this in the discussion on P23 lines 434-441:

“First, we found no evidence for a general decline in typewriting performance with day-of-week, given that backspace use remains stable on Wednesday, Thursday and Friday, and the interkey interval decreases, reflecting an increase in typing speed, from Wednesday to Friday. Second, we found that the effects of prolonged task performance on typing speed and accuracy followed a similar pattern over the different days of the week, suggesting that mental fatigue elicited on the previous day, as reflected in the effects of time-on-task and time-of-day, did not influence the course of performance during prolonged task performance on the next day. These results provide proof that mental fatigue does not accumulate across the days of the week.”

line 329 (discussion): typo "lead", should read "led"

line 333 (discussion): typo "subtitle", should read "subtle"

We corrected the typo’s in the revised manuscript.

line 336-337 (discussion): please explain in greater detail how the presented results point to information about the amount of hours employees can or should work

We agree with the reviewer that the discussion could use greater detail on the implications of the presented results. We added a paragraph in the discussion (page 25, lines 461-483):

“This study has implications for real-life working environments, given that a large part of the working population regularly performs computer work. In the Netherlands, for example, 40% of the employees perform computer work more than 6 h every day (Hooftman et al, 2019). There are several ways in which monitoring typewriting could support employees during their work. First, personalized real-time feedback based on changes in typing behavior could be provided to the users in order to help them detect when lapses in performance occur and a short break might be beneficial. However, real-time feedback might be biased due to dynamics in typewriting performance that are not related to lapses in performance. One of the characteristics of our working environment is the large variability in working conditions, due to changes in work-related tasks, noise in the working environment, and changes in general persons state, among others. Our method also allows monitoring performance over a longer period of time enabling us to detect regularities in working activities. Related to this, a second possibility of our method is to provide feedback on an individual level to help employees realize a more optimal work-break schedule that is complementary with their individual state and specific work-related demands. By comparing behavior dynamics over several weeks, typing behavior could help decide when, during the workday or -week an employee should work on tasks that need high accuracy or when it is better to work on less demanding tasks. A third option is to use changes in typing behavior to evaluate interventions in the working environment. For instance, it might provide relevant information with regard to performance efficiency for evaluating the effectiveness of a 6-hour workday instead of our regular 8-hour workday. Previously, researchers already used questionnaires to evaluate this specific intervention, however, measuring performance, and importantly, doing so without interrupting regular activities, could enhance our knowledge of its effects on performance and productivity more objectively.”

Figure 2: please note what is indicated by confidence intervals

The confidence intervals reflect the standard errors of the mean. We added this information to the figures.

---

## [Decision Letter · Decision Letter 1]

17 Sep 2020

Dynamics in typewriting performance reflect mental fatigue during real-life office work

PONE-D-20-10944R1

Dear Dr. Lorist,

We’re pleased to inform you that your manuscript has been judged scientifically suitable for publication and will be formally accepted for publication once it meets all outstanding technical requirements.

Kind regards,

Michael B. Steinborn, PhD

Academic Editor

PLOS ONE

Additional Editor Comments (optional):

Reviewers' comments:

Reviewer's Responses to Questions

**Comments to the Author**

1. If the authors have adequately addressed your comments raised in a previous round of review and you feel that this manuscript is now acceptable for publication, you may indicate that here to bypass the “Comments to the Author” section, enter your conflict of interest statement in the “Confidential to Editor” section, and submit your "Accept" recommendation.

Reviewer #1: All comments have been addressed

Reviewer #2: All comments have been addressed

2. Is the manuscript technically sound, and do the data support the conclusions?

Reviewer #1: Yes

Reviewer #2: Yes

3. Has the statistical analysis been performed appropriately and rigorously? 

Reviewer #1: Yes

Reviewer #2: Yes

4. Have the authors made all data underlying the findings in their manuscript fully available?

Reviewer #1: Yes

Reviewer #2: Yes

5. Is the manuscript presented in an intelligible fashion and written in standard English?

Reviewer #1: Yes

Reviewer #2: Yes

6. Review Comments to the Author

Reviewer #1: The authors did a good job in the revision. The manuscript improved considerably. I can recommend it in the present form.

Reviewer #2: (No Response)

7. PLOS authors have the option to publish the peer review history of their article (what does this mean?). If published, this will include your full peer review and any attached files.

Reviewer #1: No

Reviewer #2: No

---

## [Editor Report · Acceptance letter]

21 Sep 2020

PONE-D-20-10944R1 

Dynamics in typewriting performance reflect mental fatigue during real-life office work 

Dear Dr. Lorist:

I'm pleased to inform you that your manuscript has been deemed suitable for publication in PLOS ONE. Congratulations! Your manuscript is now with our production department. 

Kind regards, 

on behalf of

Dr. Michael B. Steinborn 

Academic Editor

PLOS ONE